# Retrieval Instead of Fine-tuning: A Retrieval-based Parameter Ensemble for Zero-shot Learning

## Abstract

Foundation models have become a cornerstone in deep learning, with techniques like Low-Rank Adaptation (LoRA) offering efficient fine-tuning of large models. Similarly, methods such as Retrieval-Augmented Generation (RAG), which leverage vectorized databases, have further improved model performance by grounding outputs in external information. While these approaches have demonstrated notable success, they often require extensive training or labeled data, which can limit their adaptability in resource-constrained environments. To address these challenges, we introduce Retrieval-based Parameter Ensemble (RPE), a new method that creates a vectorized database of LoRAs, enabling efficient retrieval and application of model adaptations to new tasks. RPE minimizes the need for extensive training and eliminates the requirement for labeled data, making it particularly effective for zero-shot learning. Additionally, RPE is well-suited for privacy-sensitive domains like healthcare, as it modifies model parameters without accessing raw data. When applied to tasks such as medical report generation and image segmentation, RPE not only proved effective but also surpassed supervised fine-tuning methods in certain cases, highlighting its potential to enhance both computational efficiency and privacy in deep learning applications.

## 1 Introduction

In recent years, foundation models such as CLIP (Radford et al., 2021), LLaMA (Touvron et al., 2023) and SAM (Kirillov et al., 2023) have captured significant attention for their ability to handle various tasks with minimal adaptation. Pre-trained on large datasets, these models have been successfully applied in fields such as natural language processing, computer vision, and healthcare, driving major advancements in artificial intelligence (Shu et al., 2024; Zhao et al., 2024; Rezayi et al., 2024; Yang et al., 2024).

However, fine-tuning these large models for specific tasks remains resource-intensive, often requiring substantial computational power and large-scale data. Low-Rank Adaptation (LoRA) (Hu et al., 2021) offers a solution by freezing most of the model parameters and fine-tuning only a small portion, significantly reducing the computational overhead while maintaining high performance. This is especially valuable in resource-constrained environments. Nonetheless, LoRA and similar methods are still susceptible to hallucinations—where the model generates plausible but inaccurate content—which can undermine the reliability of predictions. To address hallucination, Retrieval-Augmented Generation (RAG) (Lewis et al., 2020) incorporates an external retrieval step, grounding model outputs in factual data. Additionally, RAG excels at zero-shot learning, allowing models to handle tasks or categories without prior exposure. This is particularly important in healthcare, where models may need to recognize new diseases or interpret unfamiliar medical data with minimal labeled examples, accelerating diagnostic advancements.

Despite their strengths, fine-tuning and RAG each present significant challenges. Fine-tuning delivers superior task-specific performance but requires extensive computational and

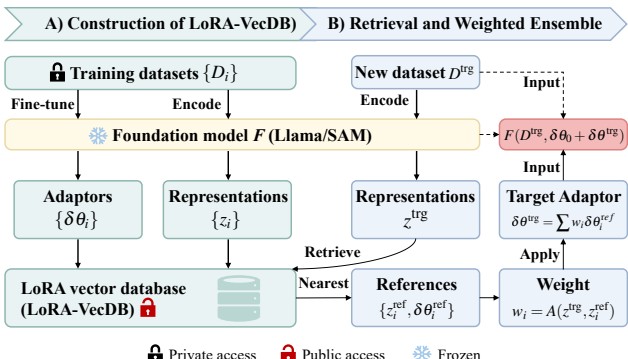

Figure 1: Pipeline of the retrieval-based parameter ensemble (RPE) model. First, a vectorized database (LoRA-VecDB) is established, containing LoRAs $\{\delta\theta_i\}$ and their representation $\{z_i\}$ across various tasks. When a new task arises, the target representation $z^{\mathrm{trg}}$ is extracted and used to query the database for similar LoRAs $\{\delta\theta_i^{\mathrm{ref}}\}$. The retrieved LoRAs are then combined using weighted ensemble methods to adapt the model to the new task without extensive fine-tuning.

data resources. RAG, on the other hand, mitigates hallucination and supports zero-shot learning but relies on access to raw data, which poses privacy concerns in fields like healthcare. Our research seeks to combine the strengths of LoRA and RAG to address both computational and privacy concerns in model adaptation. Specifically, we introduce the Retrieval-based Parameter Ensemble (RPE) model, which leverages retrieval techniques to replace traditional fine-tuning.

Our RPE model is designed to assign weights to the most relevant LoRA weights in a model ensemble. These weights are determined based on the similarity between the target task and the tasks associated with each relevant LoRA. As shown in Figure 1, the pipeline of RPE begins by establishing a vectorized database, LoRA-VecDB, for a given foundation model. This database serves as a comprehensive repository of LoRAs $\{\delta\theta_i\}$ and their corresponding representations $\{z_i\}$ across various tasks. Rather than being created by a single entity, LoRA-VecDB is a community-driven effort, promoting collaboration and ensuring the database remains accessible, diverse, and up-to-date. When a new task or dataset arises, especially in cases with limited labels or computational resources, the model's representation $z^{\mathrm{trg}}$ can be extracted and used to query LoRA-VecDB for similar adaptors $\{\delta\theta_i^{\mathrm{ref}}\}$. By calculating appropriate weights $\{w_i\}$, these LoRAs are combined to form a parameter ensemble, effectively adapting the model to the new task without the need for extensive fine-tuning.

This approach offers several key advantages. First, it significantly reduces the redundancy and computational costs typically associated with traditional fine-tuning methods. Additionally, it enhances privacy by avoiding the need to access raw data during the adaptation process. As foundation models continue to scale, the energy consumption (Samsi et al., 2023) and privacy issues (Bommasani et al., 2021) associated with their deployment become more pressing, making our RPE method a timely and valuable solution.

Our main contributions are summarized as follows:

- **Zero-shot Learning Model via LoRA Retrieval:** We introduce a pioneering zero-shot learning framework that leverages LoRA retrieval, eliminating the need for additional labeling or training, while also preserving data privacy.

- **Insights into Relationship between Parameter and Feature Spaces:** Our analysis reveals how parameter and feature spaces interact, leading to a new weighting strategy that enhances model adaptability and accuracy.

- **Real-world Validation:** We validate our approach in real-world scenarios, demonstrating its effectiveness in medical language and image processing tasks.

This paper is organized as follows: Section 2 reviews related work, providing background for our approach. Section 3 details the methodology, including the construction of the LoRA vectorized database and the retrieval process. Section 4 presents experiments evaluating the RPE model in medical applications. Sections 5 and 6 discuss the implications of our findings and suggest future research directions.

## 2 RELATED WORK

We review related work on RAG, parameter combination methods, and zero-shot learning, highlighting key advancements and differences from our approach.

**RAG** integrates external knowledge into large language models (LLMs) by retrieving relevant information to enhance generation accuracy (Ma et al., 2023). Recent advancements focus on optimizing query prompting, indexing structures, and retrieval mechanisms (Ma et al., 2023; Peng et al., 2024; Gao et al., 2022), addressing limitations of naive RAG approaches. These improvements enhance retrieval precision and reduce hallucinations in generated outputs, especially in low-resource domains. For instance, (Seo et al., 2024) leverages retrieved instances to generate new training samples with LLMs, mitigating data scarcity in specialized areas. Similarly, (Parvez et al., 2022) expands positive examples in privacy policy question-answering tasks through retriever models. However, reliance on external data introduces challenges related to privacy and computational constraints, limiting applicability in certain scenarios. For instance, some RAG methods used in LLMs retrieve raw data for the input to improve prompt quality. Others retrieve data in the feature space, which can still pose significant data privacy concerns, particularly when dealing with sensitive datasets such as those in the medical domain. In contrast, our method retrieves representations of tasks rather than specific data, ensuring the preservation of data privacy while still enabling effective task-specific adaptations.

**Parameter Combination Methods** Various methods have been developed to combine model parameters to enhance performance, robustness, and generalization. However, most current methods still require additional data for fine-tuning or additional neural network evaluations for optimization. A more detailed comparison can be found in Appendix A.1.

We aim to focus on parameter combination methods without labeled data and additional neural network evaluations. One such method is Model Soup (Wortsman et al., 2022), which simplifies model combination through parameter averaging. Another method is Federated Learning (FL) (McMahan et al., 2017), which focuses on distributed learning. In FL, multiple devices train models locally on their own data, and only parameter updates are sent to a central server, which aggregates them into a global model. This decentralized setup preserves privacy, making FL ideal for privacy-sensitive applications. FL often incorporates secure protocols and privacy-enhancing techniques, such as secret sharing (Cheng et al., 2021), to ensure data security.

**Zero-shot Learning** is a machine learning technique where a model is trained to recognize objects, categories, or concepts that it has not seen during training (Wang et al., 2019; Xian et al., 2017; Fu et al., 2018). This technique relies on the transfer of knowledge from known (seen) tasks to unknown (unseen) tasks by utilizing shared attributes or semantic relationships. In the realm of zero-shot learning, a model must from familiar tasks, denoted as $T_i^{\text{ref}}$ with corresponding parameters $\theta_i^{\text{ref}}$ to a novel task $T^{\text{trg}}$. This process requires a specific task representation $z_i^{\text{ref}}$, which is often extracted from prior knowledge sources such as textual data or structured entities. Notable studies in this field have employed neural networks to facilitate the mapping $\mathcal{A}$ from $z_i$ to $\theta_i$. For instance, DeViSE (Frome et al., 2013) used a linear mapping from image features to a joint embedding space. GCN-ZL (Wang et al., 2018) utilized Graph Neural Networks to map from word embeddings to semantic embeddings. DGP-ZL (Kampffmeyer et al., 2019) introduced Dense Graph Propagation to learn mappings from word embeddings to semantic embeddings.

Our work leverages pretrained models to obtain representations $z_i$, and replaces the traditional neural network approach with a retrieval and algorithm-based method to perform the mapping $\mathcal{A}$. This not only simplifies the generalization process but also improves the adapt-

ability of the model to new, unseen tasks. By combining advanced retrieval techniques with pretrained models, our method offers a scalable and efficient alternative to conventional zero-shot learning approaches, particularly beneficial where acquiring labeled data for all potential classes is impractical.

## 3 Method

In this section, we elaborate on two key components of our approach: the construction of the LoRA-VecDB, a vectorized database for storing model adaptations and their corresponding representations, and the retrieval and weighted ensemble mechanism. This mechanism utilizes the database to adapt foundation models dynamically to new tasks by transforming task data into query representations, retrieving relevant LoRAs, and calculating weights to configure a tailored model, thus enabling significant flexibility and performance in data-scarce or privacy-sensitive scenarios.

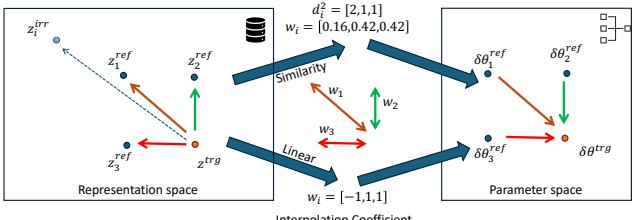

Figure 2: Workflow of retrieval and weighted ensemble stage: (1) transforming the dataset for the new task into the query representation $z^{\mathrm{trg}}$; (2) retrieving relevant LoRAs, including $\{z_i^{\mathrm{ref}}\}$ and $\{\delta\theta_i^{\mathrm{ref}}\}$; (3) computing weights $w_i$ based on the similarity between $z^{\mathrm{trg}}$ and $\{z_i^{\mathrm{ref}}\}$ in the representation space; (4) applying these weights in the parameter space to adjust $\delta\theta_i^{\mathrm{trg}}$.

### 3.1 Construction of LoRA-VecDB

The vectorized database, named LoRA-VecDB, stands as a central repository that catalogs LoRAs $\{\delta\theta_i\}$ and their corresponding representations $\{z_i\}$ for various tasks. This database not only facilitates accessibility but also encourages ongoing contributions from the community, maintaining a collaborative and up-to-date resource.

For each specific dataset $D_i$, a LoRA $\delta\theta_i$ is trained using the foundation model $F(\cdot, \theta_0)$. LoRA achieves this by freezing the pre-trained model weights and introducing trainable low-rank matrices into each layer, significantly reducing the number of parameters required for adaptation. This process also generates a representation $z_i$, capturing the essential features or transformations unique to $D_i$. Typically, the representation $z_i$ is derived directly from the feature map of $F$'s encoder, maintaining a raw projection of data features. However, for enhanced interpretability and to manage multiple adaptations, an additional encoder can be employed to refine these features into a more contextually appropriate form. This strategy draws from techniques such as RAG, where specialized encoders are employed to effectively handle large datasets.

In our application, unless explicitly stated, we utilize the feature map output from the encoder of $F$, denoted as $E^F(x_j, \theta_0)$, for individual data items $x_j$, which may represent an image or a document. This approach aligns with the strategy used in the encoder component of the MoE, where feature maps serve a pivotal role in the model architecture. It is crucial to emphasize that these feature maps are utilized in their original form, without any fine-tuning, ensuring that the integrity and the originality of the model's initial pre-training are maintained.

For simplicity and practicality in representing dataset features, we initially explored using various distribution distance metrics, such as the Chamfer distance (Borgefors, 1986),

Nearest Neighbor Distance (Alt & Godau, 1995), Mean Distance (Carroll & Arabie, 1998), to measure similarities between datasets. However, these metrics did not show significant differences in dataset characteristics. Therefore, to streamline our approach, we represent the features of dataset $D_i$ by averaging all associated data feature maps:

$$z_i = \frac{1}{|D_i|} \sum_{x_j \in D_i} E^F(x_j, \theta_0), \tag{1}$$

where $|D_i|$ denotes the number of elements in dataset $D_i$, ensuring each dataset's characteristic is represented as the mean of its features. This method not only simplifies the computational process but also facilitates the efficient storage of these averaged features in the VecDB, maintaining the integrity and accessibility of the original data representation.

Through these methodologies, LoRA-VecDB not only provides a structured and efficient way to store and retrieve adaptations but also supports a scalable framework for experimentation and enhancement in model adaptability. This open and maintained database promises to be a valuable asset for researchers and practitioners aiming to leverage existing foundation models to new datasets and problems.

### 3.2 Retrieval and Weighted Ensemble

The process begins by transforming the dataset for the new task into a query representation $z^{\mathrm{trg}}$. We then search for the most relevant LoRAs, retrieving a set of $\{z_i^{\mathrm{ref}}\}$ and $\{\delta\theta_i^{\mathrm{ref}}\}$. The weights $\{w_i\}$ are computed as a function of $z^{\mathrm{trg}}$ and $\{z_i^{\mathrm{ref}}\}$, enabling the model to utilize $F(\cdot, \theta_0 + \sum w_i \delta\theta_i^{\mathrm{ref}})$, where $\theta_0$ represents the parameters of the foundational model and $w_i \delta\theta_i^{\mathrm{ref}}$ are the weighted adjustments from the retrieved LoRAs. This methodology supports dynamic adaptation of foundational models to new tasks, leveraging community-generated adaptations and sophisticated retrieval techniques to enhance model performance without extensive retraining. The algorithm is detailed in Algorithm 1.

---

**Algorithm 1** Retrieval and Weighted Ensemble

---

**Require:** Foundation model $F(\cdot, \theta_0)$, LoRA-VecDB $\{\delta\theta, z_i\}$, target dataset $D^{\mathrm{trg}}$
**Ensure:** $F(\cdot, \theta^{\mathrm{trg}})$
1: $z^{\mathrm{trg}} = \frac{1}{|D^{\mathrm{trg}}|} \sum_{x_j \in D^{\mathrm{trg}}} E^F(x_j, \theta_0)$        ▷ Compute feature representation for $D^{\mathrm{trg}}$
2: $\{z_i^{\mathrm{ref}}\}_{i=1}^k = \mathrm{argsort}(d(z_i, z^{\mathrm{trg}}), k)$        ▷ Using $k-$NN retrieve closest LoRAs
3: $\{w_i\} = \mathcal{A}(\{z_i^{\mathrm{ref}}\}, z^{\mathrm{trg}})$        ▷ Compute weights
4: $\theta^{\mathrm{trg}} = \theta_0 + \sum w_i \delta\theta_i^{\mathrm{ref}}$        ▷ Parameter Ensemble

---

In the subsequent section, we introduce various strategies, denoted as $\mathcal{A}$, to calculate the most effective parameter inter-relationships based on latent space structures. Our findings suggest that transferring a learned LoRA from one dataset to another becomes more effective as the similarity between the datasets increases. For a clear and visual reference, please see Figure 3.

Further, we hypothesize that specific correspondences between data representations and optimal parameters allow our methods to deduce relationships between $\delta\theta_i$ based on the relationships among $z_i$. The assumptions made about the connections between the representation space and the parameter space significantly influence the derivation of different $\mathcal{A}$. This understanding aids in tailoring the algorithms to better capture and leverage these relationships, enhancing the model's performance across varied datasets.

**Similarity Calculation:** The strategy is premised on the assumption that tasks with similar feature representations are likely to benefit from similar parameter adjustments. This approach is rooted in the concept of transfer learning, where knowledge from one domain is leveraged to enhance performance in another domain. The strategy calculates the similarity between the target feature vector $z^{\mathrm{trg}}$ and and each reference feature vector $z_i^{\mathrm{ref}}$ stored in VecDB using the squared $\ell_2$ norm:

$$d^2(z_i, z^{\mathrm{trg}}) = \|z_i - z^{\mathrm{trg}}\|_2^2. \tag{2}$$

Weights are then assigned using a softmax function, which normalizes the inverse of these distances:

$$w_i = \frac{\exp(-\lambda_1 d_i^2)}{\sum_j \exp(-\lambda_1 d_j^2)}, \tag{3}$$

where $\lambda_1$ is a temperature parameter that controls the sharpness of the distribution, allowing the model to emphasize more similar LoRAs.

**Linear Combination:** The strategy is based on the assumption that a linear relationship exists between the latent representations and their corresponding parameter adjustments. This method seeks to find a linear combination of the retrieved LoRAs that best approximates the target representation, under the constraint that the combination of weights equals one, thus maintaining a normalized contribution from each LoRA.

The objective is to minimize the error between the target representation and a weighted sum of reference representations:

$$w_i = \arg\min_{\sum w_i = 1} \|z^{\mathrm{trg}} - \sum w_i z_i^{\mathrm{ref}}\|_2^2. \tag{4}$$

This optimization problem ensures that the combined parameter adjustments from the retrieved LoRAs closely match the target task's requirements.

**Regularization:** Regularization is introduced into the ensemble method to manage the influence of each LoRA, particularly when dealing with sparse or high-dimensional data. The regularization term penalizes the weights, encouraging the model to prefer simpler solutions that may generalize better. This method assumes that in the presence of many possible solutions, a sparse solution (in terms of few non-zero weights) could lead to better performance and interpretability.

The regularization strategy incorporates an $\ell_1$ norm penalty to encourage sparsity among the weights:

$$w_i = \arg\min_{\sum w_i = 1} \|z^{\mathrm{trg}} - \sum w_i z_i^{\mathrm{ref}}\|_2^2 + \lambda_2 \|w_i\|_1, \tag{5}$$

where $\lambda_2$ is the regularization parameter that balances the trade-off between the fidelity of the approximation and the sparsity of the solution. This approach is particularly useful when the number of potential LoRAs (parameters) is large, and only a subset is truly relevant for the target task.

Figure 2 illustrates demos of these methods, highlighting how similarity calculation focuses on proximity relationships with positive coefficients, while linear combination can include structural information and potentially negative coefficients. The experimental section will showcase the distinct advantages of each method.

## 4 EXPERIMENTS

### 4.1 IMPLEMENTATION DETAIL

To validate our approach, we conduct experiments using two foundational models: Llama 3.1 8B (Dubey et al., 2024) and SAM (Kirillov et al., 2023). We use 8 H100 80G GPUs for the training and fine-tuning.

For Llama 3.1 8B model, we evaluate its performance on generating medical report impressions from provided findings. Specifically, we fine tune four LoRA models derived from the pre-trained Llama 3.1 8B model using four distinct datasets collected from Massachusetts General Hospital (MGH). These datasets comprise 24,801 CT abdomen reports, 63,745 CT head reports, 18,157 MR image reports, and 60,000 X-ray image reports. Each report includes detailed image findings and corresponding impressions. We create 20 different instructions asking for impressions and remove all the names in the reports by using regular expression. The fine-tuning process employ consistent hyperparameter settings: **training batch size** = 8, **gradient accumulation steps** = 4, **optimizer** = paged adamw 32bit, **learning rate** = $5 * 10^{-6}$, **weight decay** = 0.001, **maximum gradient normal** = 0.3,

**LoRA r** = 16, **LoRA alpha** = 0.05. The number of **training epochs** is set as follows: 2 for CT abdomen, 1 for CT head, 3 for MR, and 1 for X-ray reports. In testing, we collecte 200 new reports for each type of medical image.

For SAM model, we focus on medical image segmentation tasks. Consistent with the MA-SAM framework (Chen et al., 2023), we use the same hyperparameter settings. We reproduce and train six individual MA-SAM models, each corresponding to one prostate dataset (Liu et al., 2020) that the original MA-SAM applies. For both tasks, each dataset is iteratively treated as the target dataset, while the remaining datasets serve as reference datasets for zero-shot learning. In all experiments, $\lambda_1$ in Eq 3 is set to 1 by default, and $\lambda_2$ in Eq 5 is set to 100 by default.

### 4.2 Medical report impression

We form ensemble models for each type of medical report by utilizing both similarity calculation and linear combination but without regularization. Following (Shi et al., 2024), we apply ROUGE-L (Lin, 2004), BertScore (Zhang et al., 2019) and GPT score defined in (Shi et al., 2024) in our evaluation to have a comprehensive observation for both fundamental word matching and semantic level accuracy.

| Metrics | Pre-trained | SFT | Zero-shot | | |
| --- | --- | --- | --- | --- | --- |
| | | | AVG | Ours (sim) | Ours (lin) |
| ROUGE-L | 0.1264 | 0.1387 | 0.1369 | 0.1374 | **0.1393** |
| BertScore Precision | 0.7779 | 0.7789 | 0.7811 | 0.7815 | **0.7816** |
| BertScore Recall | 0.8321 | 0.8355 | 0.8348 | 0.835 | **0.8358** |
| BertScore F1 | 0.8039 | 0.806 | 0.8068 | 0.8071 | **0.8076** |
| GPT score | 2.89 | 3.215 | **3.36** | 3.095 | 3.285 |

Table 1: Performance comparison of our models against pre-trained Llama 3.1 8B, LoRA Supervised Fine-tuning (SFT), and zero-shot models on CT abdomen medical report impression task. AVG: average ensemble, sim: similarity combination, lin: linear combination. The best values are highlighted in bold, and the second-best values are underlined.

| | CT (head) | MR | XR |
| --- | --- | --- | --- |
| Ours (sim) | 0.34 | 0.33 | 0.33 |
| Ours (lin) | 0.80 | 0.18 | 0.02 |

Table 2: Comparison of weight distributions in our similarity-based and linear combination methods for CT abdomen medical report impression task.

As shown in Table 1, we compare our models against the pre-trained Llama 3.1 8B which is the general model without additional training data, LoRA Supervised Fine-tuning (SFT) on corresponding MGH dataset, and zero-shot model that is only fine tuned on other three MGH datasets separately with average parameter ensemble. Our linear combination model achieves the best performance on CT abdomen reports across most metrics, even surpassing the SFT method. The similarity-based ensemble model also demonstrates competitive performance compared to the SFT model, which is significantly better than zero-shot pre-trained model. These results highlight that our zero-shot learning framework is not only competitive but can also outperform traditional SFT approaches in some cases. From Table 2, we observe that the similarity ensemble's weight has slightly difference from the average ensemble while surpassing it in all metrics except for the GPT score. We hypothesize that GPT may favor the average ensemble's responses, as this trend is consistent in other cases (refer to Appendix A.2), where only the GPT score is higher while other evaluation metrics are significantly lower compared to SFT and our methods. Regarding the linear combination weights, our model integerates 80% weight from the CT head model and 18% from the MR model, which is reasonable given that CT head reports share a similar pattern with CT

abdomen reports. The model also leverages knowledge from MR reports, contributing to the overall performance improvement.

### 4.3 MEDICAL IMAGE SEGMENTATION

We initiated our experiments by training LoRAs on six distinct datasets sourced from various manufacturers, each differing significantly in signal strength and resolution. This diversity introduced notable shifts in data distribution, which posed significant challenges for a single LoRA model, underscoring the necessity of training models on similar datasets to enhance task performance. For an in-depth analysis of the datasets and specific numerical evaluations, please refer to Appendix A.3.

To evaluate the efficacy of our methodology, we investigated the correlation between the similarity of datasets and the accuracy of LoRA models. Figure 3 illustrates this relationship. On the left side of the figure, each row ranks the similarity of a testing set to various training sets, with higher rankings indicating greater similarity. Correspondingly, the right side of the figure displays the accuracy rankings of LoRA models when applied to these testing sets, where higher rankings denote better performance. This visual representation confirms our hypothesis: testing sets more similar to the training sets tend to achieve higher accuracy in LoRA applications, substantiating the significant impact of dataset characteristics on model performance.

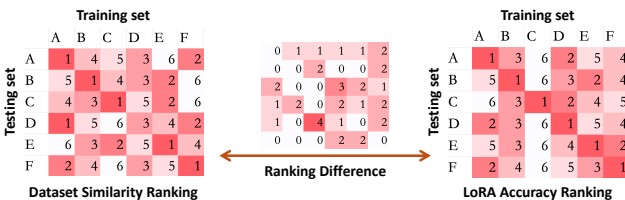

Figure 3: Correlation between dataset similarity rankings and LoRA model accuracy. The left side ranks the similarity of testing sets to various training sets, while the right side ranks the corresponding LoRA model accuracy. Higher rankings indicate greater similarity and better accuracy, respectively.

Adopting a similar approach to our medical report impression task, we computed the similarity between datasets and adjusted the LoRA representations through linear combinations, both with and without regularization, to optimize model performance for each dataset. We evaluated the effectiveness of these models using the DICE Score, a common metric for segmentation accuracy. The DICE Score is calculated as $\text{DICE} = \frac{2 \times |X \cap Y|}{|X| + |Y|}$, where $X$ denotes the set of pixels in the predicted segmentation and $Y$ denotes the set of pixels in the ground truth segmentation. The outcomes are presented in Table 3.

The pre-trained SAM model without LoRA failed to produce meaningful results. This ineffectiveness is attributed to the absence of LoRA, which deprived the model of the task-specific information necessary for accurate organ segmentation. For a detailed analysis, please refer to Appendix A.4. Our findings reveal that models employing regularized linear combinations, denoted as Ours (lin+R), significantly outperformed other methods, achieving results comparable to supervised fine-tuning.

To better understand this phenomenon, we analyzed the weights derived from different methods, focusing on testing set E as an example, detailed in Table 4. It is evident that testing set E significantly differs from the other datasets. Relying solely on similarity may not be representative. Linear interpolation without regularization results in weights that deviate significantly from the trained LoRAs, leading to suboptimal performance. Employing regularized linear combinations effectively addresses the challenges posed by significant distribution shifts in the testing set, thereby enhancing robustness and overall performance.

| Dataset | Pre-trained | SFT | Zero-shot | | | |
|---|---|---|---|---|---|---|
| | | | AVG | Ours (sim) | Ours (lin) | Ours (lin+R) |
| A | - | **95.4%** | 80.3% | 87.8% | 86.3% | 90.5% |
| B | - | **92.8%** | 77.5% | 85.0% | 83.4% | 86.0% |
| C | - | **90.5%** | 51.0% | 59.8% | 61.9% | 64.7% |
| D | - | **91.2%** | 74.9% | 82.6% | 86.7% | 90.3% |
| E | - | **92.7%** | 64.6% | 56.9% | 52.0% | 79.1% |
| F | - | **93.0%** | 82.2% | 80.8% | 82.4% | 90.3% |

Table 3: Comparison of DICE scores for our models across different testing sets against pre-trained SAM, LoRA Supervised Fine-tuning (SFT), and zero-shot models on the medical image segmentation task. AVG: average ensemble, sim: similarity combination, lin: linear combination, lin+R: regularized linear combination. The best values are highlighted in bold, and the second-best values are underlined.

| | A | B | C | D | F |
|---|---|---|---|---|---|
| Ours (sim) | 0.06 | 0.31 | 0.44 | 0.08 | 0.12 |
| Ours (lin) | -1.13 | 0.67 | 0.26 | 0.11 | 1.09 |
| Ours (lin+R) | -0.49 | 0.47 | 0.06 | -0.03 | 0.99 |

Table 4: Weight distribution of our methods applied to testing dataset E, with columns representing reference datasets for the medical image segmentation task. sim: similarity combination, lin: linear combination, lin+R: regularized linear combination.

### 4.4 Ablation Study

In this section, we present a series of ablation studies aimed at evaluating the efficacy of using the nearest LoRA compared to an ensemble approach. Additionally, we explore the potential benefits of incorporating LoRAs derived from multiple training sets in enhancing the performance of models developed through Supervised Fine-Tuning.

#### 4.4.1 Nearest LoRA vs. Ensemble Methods

A natural concern arises regarding whether it is more effective to use a model trained on the most similar dataset directly, or to employ a fusion of parameters. In this context, we explore a boundary scenario where we select only the nearest dataset's LoRA during retrieval, effectively setting $k = 1$ in a k-NN search.

Results from different datasets displayed in Table 5 reveal that relying solely on the most similar training set exhibit highly variable outcomes. Compared to the ensemble approach, using a single model tends to result in overfitting to the specific dataset it was trained on. For a more detailed discussion and numerical analysis, please refer to Table 15 in Appendix A.3. This suggests that integrating multiple models might provide a more robust and stable performance across diverse datasets.

#### 4.4.2 Whether to Improve SFT

Our model is capable of performing zero-shot learning and also serves as a method to enhance SFT. This approach proves particularly effective in scenarios where there is a shift in data distribution between the training and testing datasets, outperforming the original LoRA in certain tasks and data contexts.

Table 6 illustrates an example where ensemble coefficients are derived using all LoRA (including C's training set) variants on dataset C's testing set using linear combination. This reflects the inter-dataset relationships; notably, a negative correlation exists between the testing set of dataset C and the training set of dataset A. Using these weights, we achieved a performance of 90.8%, which slightly surpasses the 90.5% achieved by SFT. Although the

| | A | B | C | D | E | F |
|---|---|---|---|---|---|---|
| top-1 similar | 90.5% | 86.4% | 54.6% | 90.0% | 0.1% | 91.0% |

Table 5: DICE scores for different testing datasets obtained using the nearest LoRA.

| | A | B | C | D | E | F |
|---|---|---|---|---|---|---|
| weight | -0.21 | -0.07 | 1.10 | 0.05 | 0.03 | 0.11 |

Table 6: Weight distribution of linear combination including Supervised Fine-tuning LoRA applied to testing dataset C.

improvement is marginal, it suggests potential for further enhancing SFT methods, marking a promising direction for future research.

### 4.4.3 Training Cost Comparision

Fine-tuning each LoRA model for the medical report task requires around half an hour, while fine-tuning for more complex tasks and models can take several hours. In contrast, our RPE model ensembles the most relevant models for a target task in just a few minutes, making it significantly faster and more efficient. This efficiency extends to medical image segmentation tasks, where fine-tuning traditionally demands extensive computational time, but RPE achieves comparable results in a fraction of the time.

## 5 Discussion and Future work

From the experiments, it is evident that our approach yields promising results. An overall analysis based on the experimental section reveals that the RPE model significantly enhances the adaptability and efficiency of foundational models in tasks where labeled data is scarce or unavailable.

However, there are still some limitations to consider. Due to the limited number of LoRAs available, some aspects of our architecture merit further discussion. One such aspect is the potential for improving the encoder used to derive the representation $z$. This could involve utilizing a pre-trained model or specifically training an encoder to optimize weight determination. Another challenge arises when there is a large pool of LoRAs: how to efficiently retrieve and compute weights. This may necessitate further compression of both $z$ and the LoRAs themselves, although such explorations exceed the scope of this paper. This issue presents a valuable direction for future work, where enhancing the scalability and efficiency of retrieval processes could open new avenues for the application of retrieval-based machine learning models.

These insights pave the way for improving the model's robustness and applicability, particularly in privacy-sensitive or resource-constrained environments. Future research could focus on refining these aspects to fully leverage the potential of retrieval-based learning systems in broader and more diverse settings.

## 6 Conclusion

We have introduced a RPE model that achieves zero-shot learning without the need for additional data and training, while also maintaining data privacy. This model has produced promising results in medical application scenarios. Such a paradigm significantly reduces the redundant computational resource consumption of community groups and holds the potential to become an important framework in the future.

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

## A  APPENDIX

### A.1  COMPARE WITH PARAMETER ENSEMBLE METHODS

Parameter Ensemble Methods, particularly LoRA ensembles, can be categorized into three distinct types based on the requirement for labeled data and neural network evaluation: Fine-tuning, zero-shot with Neural Network Evaluation (NNE), and zero-shot without NNE.

Fine-tuning is applicable where labeled data are available for new tasks. In such scenarios, it is possible to learn coefficients for parameter combinations. For example, Zoo-Tuning (Shu et al., 2021) adapts the parameters of pretrained models to target tasks adaptively. Similarly, Mixture of Experts (MoE) methods (Xue et al., 2024; Lin et al., 2024) determine the combination weights of sub-models, termed as "routers". MoE architectures utilize a gating network to direct inputs to specialized sub-models, or "experts", that are tailored for specific tasks.

Zero-shot learning is pertinent when no labels are available for new tasks. Within this category, some methods still necessitate extensive neural network evaluations, often employing consistency regularization to enhance network performance. For instance, AdaMix (Wang et al., 2022) uses stochastic routing and consistency regularization during the training phase. UP-RLHF (Zhai et al., 2023) optimizes weights using reinforcement learning, and IOP-FL (Jiang et al., 2023) employs a consistency loss for weight optimization.

Particularly, as the computational costs of foundation models increase, zero-shot learning without NNE becomes essential in contexts lacking both labels and computational resources. Model Soup (Wortsman et al., 2022) simplifies model combination through parameter averaging. While AdaMix (Wang et al., 2022) and LoRA-Ensemble (Halbheer et al., 2024) also employ averaging during the inference phase, their contributions are often focused on other aspects. In contrast, our proposed RPE model doesn't require fine-tuning but achieve competitive performance. Unlike zero-shot with NNE, RPE doesn't optimize sub-networks or object functions. We are a zero-shot method without NNE. However, our model appliesadvanced algorithms instead of simple average. Notice that our model utilizes several datasets for ensemble while Model Soup, AdaMix and LoRA-Ensemble focus on one dataset. Our research specifically addresses the ensemble weights among different models, highlighting a unique perspective on model integration.

### A.2  OTHER EXPERIMENTS ON MEDICAL REPORT IMPRESSION

Table 7 to Table 12 shows the experiment results in other three types of medical image and corresponding weight in our methods. Our results indicate that the SFT model consistently dominates across most metrics in all experiments, with our proposed methods following closely behind. Both of our approaches significantly outperform the zero-shot pre-trained Llama 3.1 8B model, demonstrating the effectiveness of our designs. Furthermore, we observe that by making slight adjustments to the weights, the similarity ensemble model can surpass the average ensemble model in performance. Overall, our two methods are stable and consistently outperform other zero-shot approaches, showing competitiveness even against the SFT model.

### A.3  COMPARISON OF WEIGHT DISTRIBUTIONS IN OUR SIMILARITY-BASED AND LINEAR COMBINATION METHODS FOR X-RAY MEDICAL REPORT IMPRESSION TASK.

Table 13 illustrates the variability among different data sources used in our experiments. The MR datasets differ significantly in terms of strength, resolution, and manufacturer, leading to notable shifts in data distribution. Using a single LoRA for segmentation tasks tends to result in overfitting to specific data distributions and fails to generalize across diverse datasets.

To quantify the impact of these distribution shifts, we analyzed the Euclidean distances $||z_i - z_j||_2^2$ between different training and testing sets, as detailed in Table 14. Each row

| Metrics | Pre-trained | SFT | Zero-shot | | |
| --- | --- | --- | --- | --- | --- |
| | | | AVG | Ours (sim) | Ours (lin) |
| ROUGE-L | 0.201 | **0.2477** | 0.2124 | 0.2161 | 0.214 |
| BertScore Precision | 0.8166 | **0.8278** | 0.8194 | 0.8202 | 0.8201 |
| BertScore Recall | 0.8625 | **0.8739** | 0.8617 | 0.864 | 0.8629 |
| BertScore F1 | 0.8387 | **0.8499** | 0.8397 | 0.8412 | 0.8405 |
| GPT score | 4.021 | **4.735** | 4.725 | 4.27 | 4.237 |

Table 7: Performance comparison of our models against pre-trained Llama 3.1 8B, LoRA Supervised Fine-tuning (SFT), and zero-shot models on CT head medical report impression task.

| | CT (abdomen) | MR | XR |
| --- | --- | --- | --- |
| Ours (sim) | 0.32 | 0.32 | 0.36 |
| Ours (lin) | 0.25 | 0.33 | 0.42 |

Table 8: Comparison of weight distributions in our similarity-based and linear combination methods for CT head medical report impression task.

in this table shows how one testing set differs from other training sets. Correspondingly, Table 15 displays the DICE scores achieved when applying models trained on these various sets to a given testing set. The results highlight the challenges posed by dataset variability and underscore the necessity for adaptive segmentation strategies that can effectively handle diverse data characteristics.

## A.4 SAM WITHOUT LoRA

The implementation of the SAM without LoRA was found to be ineffective, as SAM lacked the necessary guidance on which organs should be segmented. As illustrated in the examples shown in Figure 4, the organs targeted by SAM for segmentation appeared to be selected randomly. In contrast, LoRAs inherently contain task-specific information, such as the identification of the organs that need to be segmented.

Despite the presence of distribution shifts across different datasets, the organ categories required for segmentation remain consistent. This consistency is crucial, as it underlines why employing LoRA enables the completion of tasks that pre-trained models without retrieval capabilities fail to achieve. This finding demonstrates the importance of integrating task-specific knowledge in the form of LoRAs to guide the segmentation process effectively, particularly when dealing with diverse medical imaging datasets.

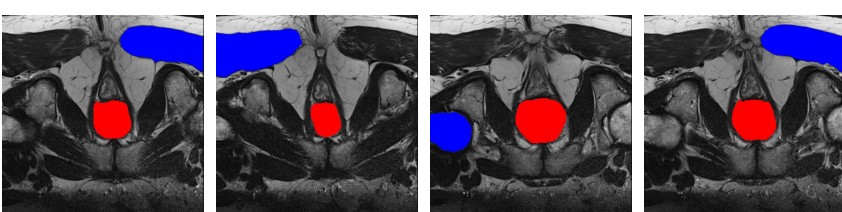

Figure 4: Pre-trained SAM segmentation outputs without the use of LoRA. The blue regions represent the segmentation results produced by SAM, while the red regions indicate the ground truth labels. This figure illustrates the randomness in organ selection by SAM when it lacks LoRA's task-specific guidance, highlighting the necessity of employing LoRA to ensure accurate and consistent organ segmentation across varying datasets.

| Metrics | Pre-trained | SFT | Zero-shot | | |
| --- | --- | --- | --- | --- | --- |
| | | | AVG | Ours (sim) | Ours (lin) |
| ROUGE-L | 0.1831 | **0.2153** | 0.1867 | 0.1914 | 0.1949 |
| BertScore Precision | 0.8107 | **0.8186** | 0.8128 | 0.8109 | 0.811 |
| BertScore Recall | 0.8644 | 0.8669 | 0.8649 | 0.8651 | **0.8671** |
| BertScore F1 | 0.8365 | **0.8418** | 0.8378 | 0.8369 | 0.838 |
| GPT score | 4.255 | 4.655 | **4.85** | 4.285 | 4.41 |

Table 9: Performance comparison of our models against pre-trained Llama 3.1 8B, LoRA Supervised Fine-tuning (SFT), and zero-shot models on MR medical report impression task.

| | CT (abdomen) | CT (head) | XR |
| --- | --- | --- | --- |
| Ours (sim) | 0.33 | 0.34 | 0.33 |
| Ours (lin) | 0.79 | 0.17 | 0.04 |

Table 10: Comparison of weight distributions in our similarity-based and linear combination methods for MR medical report impression task.

| Metrics | Pre-trained | SFT | Zero-shot | | |
| --- | --- | --- | --- | --- | --- |
| | | | AVG | Ours (sim) | Ours (lin) |
| ROUGE-L | 0.1681 | **0.2159** | 0.1776 | 0.1794 | 0.1830 |
| BertScore Precision | 0.8244 | **0.837** | **0.829** | 0.8273 | 0.8289 |
| BertScore Recall | 0.8765 | **0.8807** | 0.877 | 0.8778 | 0.8774 |
| BertScore F1 | 0.8494 | **0.858** | 0.8521 | 0.8515 | 0.8522 |
| GPT score | 4.025 | 4.845 | **4.97** | 4.17 | 4.125 |

Table 11: Performance comparison of our models against pre-trained Llama 3.1 8B, LoRA Supervised Fine-tuning (SFT), and zero-shot models on X-ray medical report impression task.

| | CT (abdomen) | CT (head) | MR |
| --- | --- | --- | --- |
| Ours (sim) | 0.32 | 0.37 | 0.31 |
| Ours (lin) | 0.48 | 0.15 | 0.38 |

Table 12: Comparison of weight distributions in our similarity-based and linear combination methods for X-ray medical report impression task.

| Dataset | Institution | Case | strength(T) | Resolution (mm) | Endorectal Coil | Manufactor |
| --- | --- | --- | --- | --- | --- | --- |
| Site A | RUNMC | 30 | 3 | 0.6-0.625/3.6-4 | Surface | Siemens |
| Site B | BMC | 30 | 1.5 | 0.4/3 | Endorectal | Philips |
| Site C | HCRUDB | 19 | 3 | 0.67-0.79/1.25 | No | Siemens |
| Site D | UCL | 13 | 1.5 and 3 | 0.325-0.625/3-3.6 | No | Siemens |
| Site E | BIDMC | 12 | 3 | 0.25/2.2-3 | Endorectal | GE |
| Site F | HK | 12 | 1.5 | 0.625/3.6 | Endorectal | Siemens |

Table 13: Characteristics of MRI datasets from multiple institutions used in the study. This table details variations in magnetic field strength, spatial resolution, usage of endorectal coils, and MRI equipment manufacturers across six different sites, highlighting the diversity of data sources in our experiments.

|   | A | B | C | D | E | F |
|---|---|---|---|---|---|---|
| A | **85.0758** | 94.8546 | 95.2915 | 89.6767 | 95.514 | 87.1439 |
| B | 97.6358 | **89.4471** | 96.5984 | 95.2394 | 90.5619 | 98.885 |
| C | 97.2556 | 97.017 | **85.7178** | 97.4879 | 95.1096 | 98.1607 |
| D | **90.9688** | 94.3976 | 96.8321 | 92.7221 | 94.3723 | 92.5209 |
| E | 101.5153 | 96.0675 | 94.905 | 100.7156 | **82.0723** | 99.3386 |
| F | 87.463 | 93.2918 | 95.6369 | 91.5755 | 95.074 | **86.0333** |

Table 14: Euclidean distances between feature vectors of different datasets, quantifying the distribution shifts. Each entry represents the squared Euclidean distance $||z_i - z_j||_2^2$ between testing sets and trainig sets across sites A through F. The closest distances are highlighted in bold.

|   | A | B | C | D | E | F |
|---|---|---|---|---|---|---|
| A | **95.4**% | 92.4% | 44.3% | 91.0% | 83.3% | 90.5% |
| B | 84.1% | **92.8**% | 44.8% | 87.0% | 86.4% | 85.3% |
| C | 26.1% | 60.2% | **90.5**% | 75.1% | 54.6% | 39.0% |
| D | 90.0% | 86.7% | 49.9% | **91.2**% | 71.5% | 76.4% |
| E | 75.5% | 84.8% | 0.1% | 76.8% | **92.7**% | 85.8% |
| F | 91.0% | 87.4% | 58.2% | 84.3% | 90.1% | **93.0**% |

Table 15: DICE scores for models tested across different datasets, reflecting model performance variability. The highest scores are highlighted in bold.

