# OpenReview forum: "Retrieval Instead of Fine-tuning: A Retrieval-based Parameter Ensemble for Zero-shot Learning"
_ICLR.cc/2025/Conference — Submitted to ICLR 2025_

### Official Review · Reviewer_FhMu · 2024-10-27

**Soundness:** 2
**Presentation:** 2
**Contribution:** 2
**Rating:** 5
**Confidence:** 3

**Summary:**

Summary:
The paper combines LoRA and parameter ensembling methods to address to adaptation of foundation models.
Contributions
1.The proposed RPE adapts the foundation model without the requirement of labeled data
2.RPE is effective to zero-shot learning
3.The paper conducted experiments on extensive medical data

**Strengths:**

1.The paper studies an interesting issue
2.The paper proposes several strategies to compute the weights
3.The paper conducts experiments on different medical datasets

**Weaknesses:**

1.The paper is not self-contained, for example, the author should explain the meaning of {\delta\theta_i}, {\delta\theta_i^{ref}} and how to obtain them.
2.The improvement of the proposed method is mainly attributed to LoRA, the improvement brought by RPE is marginal

**Questions:**

1.What is the dataset handling process?
2.How do you obtain the four LoRA models? What is the model pre-training process?
3.Please briefly describe the evaluation metrics  and the datasets used for the segmentation task

---

> ### Author Response · Authors · 2024-11-23
>
> Thank you for your detailed and insightful feedback on our manuscript. We appreciate the opportunity to address the concerns raised. Below, we provide clarifications and revisions that we believe adequately respond to each point of criticism.
>
> 1. Clarification
>
> To address this, we have revised the manuscript to include a detailed explanation of the terms ${\delta\theta_i}$ and ${\delta\theta_i^{ref}}$.
>
> 2. Dataset Handling Process
>
> The four LoRA models are fined tuned using extra data. In our paper, we use trl library from Hugging Face. For large language models, it is unrealistic to pre-train the model because it requires millions of dollars and several months to pre-train. Instead, most researches in LLMs just apply the pre-trained foundation models.
>
> 3. Evaluation Metrics
>
> In section 4.1 we introduce our datasets used for segmentation task. In section 4.3 we specify the DICE score, a common metric for segmentation accuracy.
>
> We hope that these revisions and clarifications address the concerns raised by the reviewer and strengthen the contribution of our work. Thank you for considering our rebuttal and the revised manuscript.

---

> > ### Comment · Reviewer_FhMu · 2024-11-25
> > **Thank you for the comments**
> >
> > Thank you for the comments. It helps clarify some problems. I maintain my original rating.

---

### Official Review · Reviewer_1Fmq · 2024-10-27

**Soundness:** 2
**Presentation:** 3
**Contribution:** 2
**Rating:** 5
**Confidence:** 3

**Summary:**

The paper introduces the Retrieval-based Parameter Ensemble method for Foundation Models that enables zero-shot adaption of foundation models. The key idea is to combine the LoRA adaptation with Retrieval-Augmented Generation mechanism. Basically, the authors consider K training datasets, adapt the foundation model using LoRA on each dataset and save the vectorized representation of the dataset along with the fine tuned LoRA weights on the vector database. To adapt to a new dataset, representation of the new dataset is computed, and its similarity with all the representations in the vector database is computed to obtain the weights. The weights are used in the ensemble of the saved LoRA parameters. Such proposed LoRA ensembling based fine-tuning enables the model to be adapted to new task in a zero-shot manner. Based on the experimental results, the proposed technique seems to be effective compared to single-task LoRA finetuning, and is beneficial in zero-shot learning.

**Strengths:**

- The proposed idea of Retrieval-based parameter ensemble is simple and intuitive. It is quite straightforward and can easily be applied.
- The proposed method bypasses the cost of training models on new datasets i.e. it is zero shot, and based on the empirical results, appears to be competitive. No expensive fine-tuning is required.
- Data intensive retraining is not required in the proposed approach, and the proposed approach can minimize the privacy leaks, improving on privacy for sensitive data.

**Weaknesses:**

Scalability: I think the proposed idea will face scalability issue when large number of datasets appear. For each dataset, LoRA weight, along with the dataset representation would have to be stored. The experiments only consider setting with highly limited number of datasets/LoRA adaptations (4 LoRA parameters for medical report, and 6 LoRA parameters for image segmentation). I think the scalability would be major issue with number of tasks. The empirical evaluation is limited. Retrieval efficiency and storage overhead may become an issue in real-world application with large number of tasks.

Dataset representation: Representing dataset with the mean of embeddings of each data point of the dataset is quite restrictive (Eqn. 1). Though effective on the two narrow experiments carried out, this approach may not be effective and generalize for more challenging settings and problems. Some theoretical support analysis or guarantees could significantly strengthen the work. Alternatively, comprehensive empirical analysis across a diverse range of datasets could strengthen the work.

The parameter ensemble needs to weight all the LoRA parameters that may be computationally expensive, especially when considering large number of datasets, and corresponding LoRA fine-tuned weights. Moreover, the fundamental assumption of similarity of dataset representation to similarity of lora weights in parameter space may not be true. This needs to be empirically validated on more general settings beyond specialized medical task considered in the work.

Say a completely new task/dataset appears that is distinct from the existing tasks in the vector database. Current approach is unlikely to work in such setting.

Minor typos:
Typos:  Page 3, line 159 --> from, line 344 --> reports should be report,

**Questions:**

Please see and clarify on the concerns of the weaknesses section.

---

> ### Author Response · Authors · 2024-11-23
>
> We appreciate the detailed feedback provided on our manuscript. Below, we address each point raised in the review and provide clarifications for our research.
>
> 1. Scalability Concerns
>
> We envision leveraging retrieval and compression algorithms similar to those used in RAG systems to address these challenges. While our current experiments were limited due to resource constraints, demonstrating our approach's effectiveness on a small scale, we plan to explore scalable solutions as part of our future work. Techniques such as efficient database indexing, data compression algorithms, and more advanced retrieval mechanisms will be considered to enhance scalability without compromising performance.
>
> 2. Effectiveness in More Challenging Settings
>
> We acknowledge that our approach, while effective in the constrained settings of our experiments, may face limitations in more challenging zero-shot learning scenarios. Our method does not claim to solve zero-shot learning entirely but proposes an improved strategy for weight selection that moves beyond simple averaging methods typically used. This approach is particularly aimed at enhancing performance where traditional methods fall short, providing a stepping stone towards tackling more complex zero-shot scenarios.
>
> 3. Computational Costs
>
> The computational cost of our method is predominantly influenced by two factors: the retrieval of the nearest neighbors and the optimization based on these neighbors. For the retrieval part, we can draw on established methods like those used in RAG, which are designed to handle large-scale data efficiently. The optimization process, which involves calculations based on a limited number of vectors (k vectors), has been demonstrated in Section 4.4.3 to have a very small computational cost compared to fine-tuning and inference.
>
> 4. Empirical Validation and Theoretical Support
>
> We accept the suggestion to empirically validate our assumptions regarding the similarity of dataset representations to LoRA weights in parameter space across more general settings. Future work will include extensive testing beyond specialized medical tasks to include a broader range of datasets and task types.
>
> We hope that these revisions and clarifications address the concerns raised by the reviewer and strengthen the contribution of our work. Thank you for considering our rebuttal and the revised manuscript.

---

> > ### Comment · Reviewer_1Fmq · 2024-11-27
> > **Final Decision**
> >
> > I thank the reviewers for the rebuttal to my reviews.
> >
> > Some of my concerns have been addressed. However some concerns remain. Also considering the other reviewer's concerns, I believe that the work needs some improvement before publication. I've decided to keep my score to 5 (marginally below the acceptance threshold).
> >
> > Some unaddressed comments/areas that authors could focus on to improve the work are:
> > - Validate the work by carrying out experiments in more challenging settings, for eg. by carry more thorough ablations for dataset representations, broader range of datasets and task types
> > - Clarify on some of the concerns (eg. Say a completely new task/dataset appears that is distinct from the existing tasks in the vector database. How could the work address such settings?... A potential solution could be to have a embedding space threshold to decide when to use the vector dataset and when to do LoRA adaptation based on some similarity metric... Other better solutions could also exist..
> >
> > Best of luck to the authors.

---

### Official Review · Reviewer_hzPp · 2024-11-04

**Soundness:** 2
**Presentation:** 2
**Contribution:** 3
**Rating:** 5
**Confidence:** 3

**Summary:**

This article provides an example of using the RPE (retrieval-based parameter-efficient) approach, which leverages retrieval instead of fine-tuning. The author’s work uses pre-trained models to obtain representations and replaces the traditional neural network approach with a retrieval and algorithm-based method to perform mapping.  The core contribution is the use of a k-nearest neighbors (kNN) method to retrieve the closest LoRA modules, which are then used to compute weights and incorporate regularization to further improve performance. I believe this method shows promise. However, based on the current experimental setup, I’m uncertain about the performance advantage. This article needs some revisions and should provide more evidence to demonstrate the originality and performance improvements of these methods compared to others.

**Strengths:**

The model proposed in this article is well-structured and straightforward to implement. The experimental results, closely tied to medical data, highlight the model's significant potential for practical applications in the future. According to the model description, this method does not require additional labeling and can ensure privacy. However, these advantages have not been fully validated in the current experiments.

**Weaknesses:**

However, based on the current experimental setup, I’m unsure about the performance advantage. For instance, in Table 3, there is insufficient discussion about why the ensemble method outperforms other methods. The description of the experiment lacks details, especially as the data pertains specifically to the medical field. Additionally, the author claims that this ensemble method is computationally efficient, but there is no experimental evidence to validate this efficiency.

**Questions:**

The author claims that this ensemble method is computationally efficient. If possible, could the author provide specific information on the model's time performance?

Additionally, I have a few other suggestions. The article contains a significant amount of specialized medical knowledge, so I recommend adding more background information when explaining the experiments, as many readers may not have a medical background. Given that the experiments primarily focus on medical data, I am curious whether this ensemble method is specifically suited for certain tasks in the medical field, or if it has the potential to generalize across broader applications. It would be helpful if the author could clarify this aspect.

I also encourage the author to discuss the similarities and differences between this method and other ensemble approaches that use LoRA models. For instance, is there any connection between this article and the following studies? I have concerns regarding the originality of this work, and further clarification on this point would be appreciated.

Halbheer, M., Mühlematter, D.J., Becker, A., Narnhofer, D., Aasen, H., Schindler, K., and Turkoglu, M.O., 2024. LoRA-Ensemble: Efficient Uncertainty Modelling for Self-attention Networks. arXiv preprint arXiv:2405.14438.
Zhai, Y., Zhang, H., Lei, Y., Yu, Y., Xu, K., Feng, D., Ding, B., and Wang, H., 2023. Uncertainty-penalized reinforcement learning from human feedback with diverse reward LoRA ensembles. arXiv preprint arXiv:2401.00243.
Halbheer, M., Mühlematter, D.J., Becker, A., Narnhofer, D., Aasen, H., Schindler, K., and Turkoglu, M.O., 2024. LoRA-Ensemble: Efficient Uncertainty Modelling for Self-attention Networks. arXiv preprint arXiv:2405.14438.
Finally, the article lacks certain essential details. For example, the section on regularization does not specify how to set the regularization parameter, which could hinder readers from replicating the experiments.

---

> ### Author Response · Authors · 2024-11-23
>
> Thank you for your constructive comments and suggestions regarding our manuscript. We appreciate the opportunity to clarify the concerns raised and to strengthen our paper. Below, we address each point specifically:
>
> 1. Computational Efficiency and Experimental Evidence
>
> As suggested, we have now included detailed information on the computational efficiency of our ensemble method in Section 4.4.3.
>
> 2. Applicability to Medical Data and Background Information
>
> We acknowledge the concern regarding the specialized use of medical data. Our method, while not exclusively designed for medical applications, is particularly suited to scenarios requiring stringent privacy protections, such as medical settings. To clarify this, we have revised the manuscript to include a more thorough explanation of why medical data was chosen for the experiments. Additionally, we have supplemented the paper with background information on the medical aspects discussed.
>
> 3. Comparison with Other Ensemble Methods Using LoRA Models
>
> To address the request for a clearer differentiation between our method and other ensemble approaches, particularly those that utilize LoRA models, we have added a new section in Appendix A1. This section delineates the key differences and application scenarios of our method compared to others. Most notably, it highlights that many existing methods require additional data for fine-tuning or neural network evaluation for optimization, which is not feasible in label-scarce and computationally constrained environments. In such cases, most current methods employ parameter averaging. Our approach, using a weighted average method rather than a simple average, is distinct in its efficiency and practicality under these constraints.
>
> 4. Regularization Parameter Settings
>
> We have amended the manuscript to include explicit details on how the regularization parameters were set, facilitating replication of our experiments by other researchers.
>
> We hope that these revisions and clarifications address the concerns raised by the reviewer and strengthen the contribution of our work. Thank you for considering our rebuttal and the revised manuscript.

---

> > ### Comment · Reviewer_hzPp · 2024-11-27
> > **Final Decision**
> >
> > I carefully reviewed all the revisions to the article and weigh the rebuttal. The updated version has improved the readability of the paper, enhancing its presentation to a certain extent. The authors have explained why they used the medical dataset, but some concerns remain unaddressed. In particular, I am very concerned about whether this method is generalizable, and I would have liked to see some data to demonstrate this. Besides, If the originality of the method in this article needs to be demonstrated, the current data appears to be somewhat insufficient. Therefore, while I acknowledge that the presentation of the paper has improved after the revisions, I will maintain my overall score (5) unchanged.
> >
> > Once again, I would like to thank the authors.

---

### Official Review · Reviewer_Dvgv · 2024-11-05

**Soundness:** 2
**Presentation:** 1
**Contribution:** 1
**Rating:** 3
**Confidence:** 5

**Summary:**

The paper introduces Retrieval-based Parameter Ensemble (RPE), a zero-shot learning approach that leverages Low-Rank Adaptation (LoRA) parameters stored in a vectorized database, LoRA-VecDB, to adapt large models to new tasks without fine-tuning. For each new task, RPE retrieves and combines relevant LoRA parameters from the database based on task similarity, creating a weighted ensemble. This method targets efficient, privacy-preserving model adaptation for diverse tasks.

**Strengths:**

- Efficient Parameter Retrieval: The paper proposes an approach for zero-shot learning that retrieves LoRA parameters to create task-specific adaptations, reducing the need for traditional fine-tuning.
- Privacy Consideration: The paper is motivated to tackle privacy concerns of RAG.

**Weaknesses:**

1. Misinterpretation of Privacy Concerns in RAG
- The paper inaccurately claims that retrieval-augmented generation (RAG) approaches require access to raw data, posing privacy risks. In reality, RAG systems typically retrieve from embedding databases, not raw data, and privacy-preserving variants (e.g., using federated learning or differential privacy) already exist. The paper would benefit from a more accurate representation of RAG’s privacy characteristics and should clarify how its approach offers advantages over these established privacy-preserving RAG methods.
2. Limited Novelty in Algorithmic Contribution
- The proposed approach closely resembles existing model zoo-based zero-shot learning techniques, such as Task2Vec, Zoo-Tuning, HyperSTAR, and Ada-Mix, which also retrieve and adapt pre-trained models based on task similarity. The main difference is the use of LoRA parameters instead of full models, offering storage advantages but not a fundamentally new algorithmic approach. The paper would be strengthened by explicitly differentiating its method from these works, detailing any unique technical contributions beyond storage efficiency.
3. Incomplete Task Representation and Retrieval Process
- The paper lacks clarity on how task representations for each dataset are generated and subsequently used in the retrieval process. Given that these representations are central to the model selection mechanism, the methodology would benefit from a detailed description of how task representations are created and validated, as well as a discussion on how task representation quality impacts retrieval performance.
4. Over-Reliance on Assumptions in Realistic Settings
- The paper assumes that a large pool of downstream LoRA models with well-defined task representations is readily available, but in practice, obtaining these representations at scale is both challenging and expensive. Additionally, the process of storing and accessing task representations carries its own privacy concerns when derived from potentially sensitive datasets. Addressing these feasibility and privacy challenges more thoroughly would improve the paper’s practicality and strengthen its claim of scalability.
5. Narrow Experimental Scope
- The experiments are limited to two tasks—medical image segmentation and medical report generation—both in the medical domain. This narrow scope makes it difficult to assess the generalizability of the approach across diverse domains or standard zero-shot learning benchmarks. Expanding the evaluation to include varied datasets and task types would provide stronger evidence of the method’s adaptability and effectiveness in broader applications.
In summary, addressing these issues would improve the rigor, novelty, and generalizability of the paper.

[1] Task2Vec: Task Embedding for Meta-Learning
[2] Zoo-Tuning: Adaptive Transfer from a Model Zoo
[3] HyperSTAR: Task-Aware Hyperparameters for Deep Networks
[4] AdaMix: Mixture-of-Adaptations for Parameter-efficient Model Tuning

**Questions:**

Please refer to each bullet in the weakness.

---

> ### Author Response · Authors · 2024-11-23
>
> Thank you for your detailed and insightful feedback on our manuscript. We appreciate the opportunity to address the concerns raised. Below, we provide clarifications and revisions that we believe adequately respond to each point of criticism.
>
> 1. Misinterpretation of Privacy Concerns in RAG
>
> It is correct that RAG systems typically retrieve information from embedding databases rather than raw data. However, in the context of LLMs, RAG is often employed to retrieve examples and instances as supplementary information for prompts to improve the accuracy of generated results. For instance, it may retrieve word translations and example sentences for low-resource language translation tasks. In such cases, the retrieved information needs to be converted back into raw data to serve as input for prompts, which raises data privacy concerns. However, our RAG-based algorithm does not require this step. Besides, our method compresses the retrieval dataset into a vector in high-dimensional space to represent the task rather than specific data. This vector encapsulates the biases of different tasks—such as between CT reports and MRI reports, making it extremely challenging to reconstruct any individual patient’s data from this representation alone, thereby preserving data privacy. We have now added a more detailed explanation of privacy concerns in the RAG-related works section to address this point.
>
> 2. Limited Novelty in Algorithmic Contribution
>
> For the algorithmic contribution, we have added more explanation to clearly distinguish our methods from the four referenced works in Section 4.4.3. In summary, our algorithm is both model- and task-agnostic. While the method is simple, it incurs no additional neural network evaluation during inference. The time required to compute the corresponding weights for each model is significantly shorter than fine-tuning (e.g., several minutes versus several hours), yet we can still achieve fine-tuning-level performance. Regarding Task2Vec, it involves using a “probe” network pre-trained on ImageNet as a feature extractor and retraining the classifier layer for any given task, making it neither model- (CNN) nor task- (image classification) agnostic. Retraining LLMs is also challenging due to the enormous computational resources required. This issue also applies to methods such as Zoo-Tuning, HyperSTAR, and Ada-Mix. All these approaches require network optimization, retraining, or tuning, which are not computationally resource-efficient.
>
> 3. Incomplete Task Representation and Retrieval Process
>
> For the task representations of each dataset, as mentioned in Section 3.1, we encode the dataset into the feature space and then compress it into a single high-dimensional vector using Equation 1. For a new task (or incoming data), we use the same encoder to compute its task representation. Subsequently, we calculate the corresponding weights for each model based on the task representation using similarity computation and linear combination denoted as A in Algorithm 1, as illustrated in Section 3.2.
>
> 4. Over-Reliance on Assumptions in Realistic Settings
>
> In realistic settings, there are currently thousands of LoRA weights available on Hugging Face, spanning various tasks, models, and modalities. Our method is a pioneering framework designed to effectively utilize these abundant LoRA weights. We believe that using shared LoRA instead of individuals and groups training their own LoRA will become a trend, and more and more people will contribute. Many databases used in RAG applications also come from public databases, rather than being limited to privately constructed databases.
>
> 5.Narrow Experimental Scope
>
> We acknowledge that we have applied our methods to only image segementation and impression generation. However, our methods can be easily extended to other domains and tasks. Although thousands of LoRA weights are available on Hugging Face, it is still necessary to obtain the datasets representations used to train these weights. Additionally, we aimed to verify our methods without any external interference. For this reason, we chose to fine-tune our own LoRA weights, which is time-consuming. Expanding the range of tasks will be a focus of our future work.
>
> We hope that these revisions and clarifications address the concerns raised by the reviewer and strengthen the contribution of our work. Thank you for considering our rebuttal and the revised manuscript.

---

> > ### Comment · Reviewer_Dvgv · 2024-11-25
> >
> > Thanks for the response from the authors. Yet, most of my concerns maintain:
> > 1. Novelty:
> > From my understanding, the major claim of the paper's contribution is that, given an unseen task, RFE embedds the task into a representation and does retrieval from a pool of (task representation, LoRA models) pairs from select ensembles LoRAs based on similarity score. The idea is not new, but introduced in Task2Vec, Zoo-Tuning, HyperSTAR, and Ada-Mix. The paper is more like an existing approach being adapted in a new scenario with LoRA as the model and LLM as embeddings.
> >
> > Also, using the zero-shot global mean as task presentation is mentioned in Table 1 in HyperSTAR. Due to the prior LLM era where the generalization ability of models is limited, previous methods such as Task2Vec, Zoo-Tuning, HyperSTAR, and Ada-Mix introduced a light-weight module for task representation but had baselined the zero-shot global mean. Unfortunately, I did not see any innovations along this thread.
> >
> > 2. Over-Reliance on Assumptions in Realistic Settings:
> > It is true that many LoRAs models are accessible but not the respective dataset representations. It is hard to ask for the community to upload both models and dataset representations, and without either of them, RFE will be hard to deploy, limiting its usability. I acknowledge that expanding the range of tasks will be a focus of future work, but given the current version, I don't this the experiment is sound and generalizable enough to reach the acceptance bar.

---

> > > ### Author Response · Authors · 2024-11-25
> > >
> > > Thank you for your response. Below, we provide more clarifications and revisions.
> > > 1.Novelty: Our idea has some similarity part with the four methods you mentioned. However, we also have our unique similarity score(linear combination), which can give us negative weight. This negative weight plays an important role in our model ensemble. For more details please refer to ablation study section. Besides, applying task representation and similarity calculation in LLMs has its own benefits. As we mentioned, our methods is model and task agnostic while Task2Vec, Zoo-Tuning, HyperSTAR, and Ada-Mix are not. That is why they have similar ideas but still publish their own methods based on different model structures and data. We believe we are the first work that applies this idea to LoRA weight from different datasets and it can be applied to a wide variety of models.
> > > 2.Over-Reliance on Assumptions in Realistic Settings: We can provide the code for community to calculate the dataset representations in a very shot time since our RPE method is very light weight, and we don't require a lot of models to do ensemble (usually within 10 models). It is beneficial to protect data privacy as well as make full use of their fine-tuned models to save computational resource. If the community is not able to calculate the representations, we still have many LoRA weights accompanied with corresponding fine-tune data (e.g MA-SAM in our paper). In this case, we can still calculate the data representations by using our code. In worse case, if some institutions (e.g hospitals)  can't provide data and even model weights because of data privacy concern, we can still leverage our RPE model to ensemble a competitive model from open source data and LoRA.

---

### Author Response · Authors · 2024-11-23

First and foremost, we extend our deepest gratitude for your insightful comments and constructive critiques. Your feedback has been instrumental in refining our manuscript. We have addressed each point in our revised submission, with all modifications clearly marked in red. Here, we wish to discuss common concerns raised during the review process and clarify some central aspects of our work.

1. Distinction of Our Algorithm from Other Parameter Ensemble Methods:

Parameter Ensemble Methods, particularly LoRA ensembles, can be categorized into three distinct types based on the requirement for labeled data and neural network evaluation: Fine-tuning, zero-shot with Neural Network Evaluation (NNE), and zero-shot without NNE.

Fine-tuning corresponds to scenarios where labeled data are available for new tasks. Zero-shot learning is applicable when there are no labels for new tasks. Within this, some methods still require extensive neural network evaluations, often relying on consistency regularization to optimize network performance. Zero-shot without NNE becomes crucial in situations devoid of labels and computational resources. While most current methods default to averaging approaches, our method innovatively considers task similarity, which we detail in Appendix A1. The computational costs associated with our method are thoroughly discussed in Section 4.3.3.

2. Narrow Experimental Scope & Over-Reliance on open community

The computational costs of training and fine-tuning foundational models are substantial, which is why our efforts focus on testing our algorithms on practical and limited application scenarios and models. We believe that leveraging publicly available models, rather than training private models, is a sustainable trend for the future, especially as the energy consumption of large models increases. This approach not only reduces redundancy but also minimizes wastage inherent in training private models.

3. Potential Challenges with Novel Complex Tasks

We concede that our discussion has certain limitations. It is challenging to validate our assumptions about the similarity of dataset representations to LoRA weights in parameter space across more generalized settings. This intrinsic difficulty is a known challenge within zero-shot learning. In future work, we plan to test our hypotheses across a broader array of application scenarios to better understand and refine our approach.

Thank you once again for your thorough evaluations and for aiding in the improvement of our research.

---

### Meta-Review · Area_Chair_P16c · 2024-12-13

**Metareview:**

I have read all the materials of this paper including the manuscript, appendix, comments, and response. Based on collected information from all reviewers and my personal judgment, I can make the recommendation on this paper, reject. No objection from reviewers who participated in the internal discussion was raised against the reject recommendation.

**Research Question**

The paper considers the LLM fine-tuning problem.

**Challenge Analysis**

The authors claim that the current LLM fine-tuning needs the data for new tasks.

**Philosophy**

The authors aim to solve the research question from the retrieval perspective. Concretely, the authors reuse the knowledge from existing fine-tuned tasks for the new task.

**Technique**

To implement the above idea, the authors build a database to store the existing fine-tuned tasks, and ensemble the existing ones to fit the new task. In general, the techniques are straightforward. But the technical contribution is too limited.

**Experiment**

The experimental results are not extensive and promising, due to 1) lack competitive methods in the same setting and 2) inferior performance compared to SFT. If the results are not competitive with SFT, the authors need to target a scenario where SFT fails or is not practical.

The reviewer team made the rejection recommendation due to limited novelty in techniques and unsolid experimental results. I do not see much difficulty to solve the targeted research question. In another words, I do not learn much insights from this paper.

**Additional Comments On Reviewer Discussion:**

No objection from reviewers who participated in the internal discussion was raised against the reject recommendation.

---

### Decision · Program_Chairs · 2025-01-22

Reject